# EviMix: Evidential Deep Learning with Latent-Space Mixing for Uncertainty Quantification and OOD Detection

## Abstract

Reliable uncertainty quantification (UQ) is essential for deploying deep neural networks in safety-critical domains such as autonomous driving and medical imaging. Evidential Deep Learning (EDL) provides a computationally efficient framework for estimating epistemic and aleatoric uncertainty through Dirichlet evidence assignment, enabling *real-time* uncertainty estimation. However, recent studies have raised concerns about its robustness, including conflation of uncertainty types, persistent epistemic uncertainty under abundant data, and sensitivity to training dynamics. Moreover, the interaction between EDL and modern data augmentation strategies remains poorly understood. In this work, we make three contributions: (1) we systematically study how popular pixel-space mix-based augmentations affect EDL's OOD detection and uncertainty estimates. (2) we introduce *EviMix*, a feature-space augmentation framework that interpolates latent representations across multiple network depths using both in-batch cross-class mixing and an external mixing set, with layer-wise severities that decay with depth (early layers emphasize stronger low-level perturbations, while later layers emphasize semantic interpolation). and (3) we couple these severities to the EDL objective to explicitly regulate aleatoric and epistemic uncertainty. Experiments show that *EviMix* improves OOD detection, promotes stronger functional specialization between uncertainty components, and enhances calibration compared to pixel-space and single-layer feature mixing baselines.

## 1 Introduction

Deep neural networks excel in vision, language, and robotics, but their deterministic outputs can hide unreliability under distribution shifts-posing serious risks in safety-critical fields like autonomous driving and healthcare. Overconfidence on out-of-distribution (OOD) inputs can cause catastrophic failures, making uncertainty quantification (UQ) essential for trustworthy AI.

Existing UQ methods like deep ensembles (Lakshminarayanan et al., 2017) and Bayesian neural networks (Blundell et al., 2015) achieve strong results but are computationally costly. Evidential Deep Learning (EDL) (Sensoy et al., 2018) provides a more efficient alternative by parameterizing Dirichlet distributions over class probabilities, yielding predictions and uncertainty in a single forward pass. However, its behavior under data augmentation remains largely unexplored.

Data augmentation both regularizes models and boosts robustness, but it can also distort uncertainty estimates. Pixel-space mixup methods such as MixUp (Zhang et al., 2017), CutMix (Yun et al., 2019), AugMix (Hendrycks et al., 2019), PixMix (Hendrycks et al., 2022), and the recent LayerMix (Ahmad et al., 2025) can improve calibration and OOD sensitivity, largely by injecting visual diversity (e.g., textures, patterns, and compositional shifts). However, these perturbations often primarily inflate *aleatoric-like* uncertainty and do not explicitly regulate *epistemic* behavior in the learned representation. Recent work (Juergens et al., 2024) further shows that EDL often fails to clearly separate epistemic ignorance from aleatoric noise, leaving it vulnerable under distribution shifts.

Our key insight is that *augmentation depth dictates the type of uncertainty probed*: early-layer perturbations highlight aleatoric noise, while later-layer cross-class interpolation induces epistemic

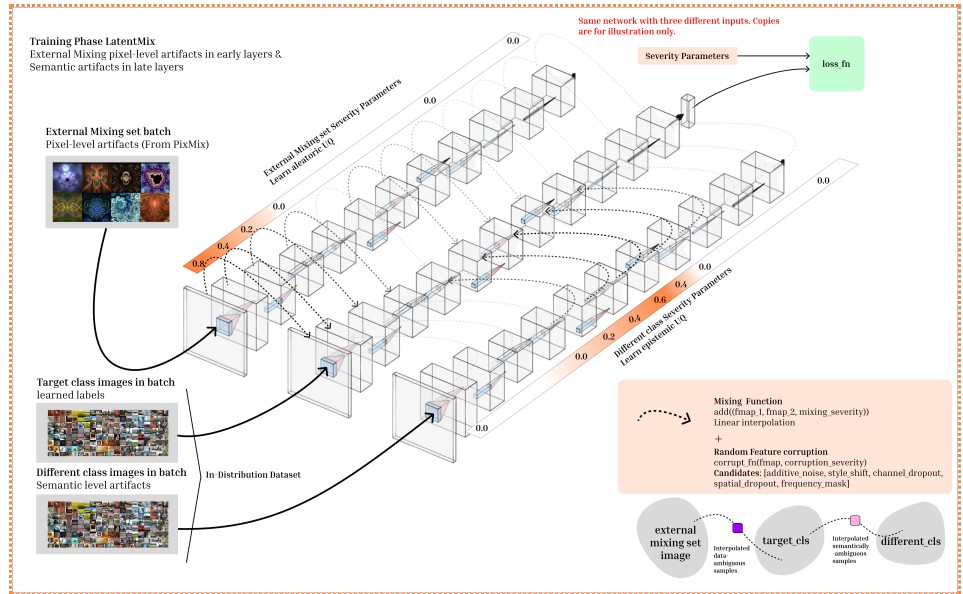

Figure 1: Illustration of the proposed **EviMix** method. The network is trained with both a standard image batch and an auxiliary mixing set. Early layers apply stronger feature mixing that decreases with depth, while in-batch cross-class mixing is emphasized in deeper layers to induce richer semantic interpolations. For each sample, a mixing severity is drawn from predefined layer-wise parameters and applied via stochastic corruptions (additive noise, style shift, channel dropout, spatial dropout, frequency masking) before linearly interpolating features from either the auxiliary set or other in-batch classes. This severity governs both feature interpolation and evidential loss regularization, shaping aleatoric and epistemic uncertainty estimates. Compared to pixel-space mix-based augmentation (e.g., PixMix (Hendrycks et al., 2022), LayerMix (Ahmad et al., 2025)) and single-layer feature-space mixing (e.g., Manifold Mixup (Verma et al., 2019)), EviMix performs structured interpolation across multiple depths with severity-aware loss coupling.

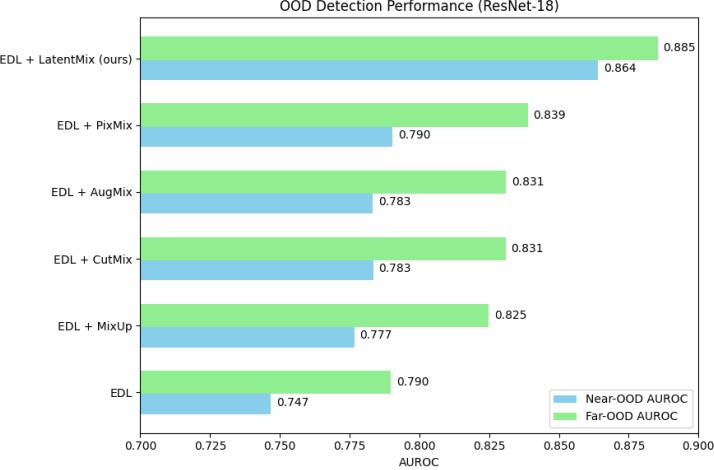

Figure 2: OOD Detection AUROC Evaluation using ResNet-18 (He et al., 2016) trained on CIFAR-10: EviMix achieves highest AUROC scores. Refer to 4 regarding OOD datasets.

uncertainty by pushing representations off-manifold. This motivates *EviMix*, which performs structured latent interpolation across multiple depths with controllable severity. By disentangling aleatoric and epistemic effects in feature space, *EviMix* offers a principled augmentation framework that strengthens evidential learning and improves reliability under distribution shift (see Figure 1).

Figure 2 provides a teaser result on CIFAR-10 OOD detection (AUROC), highlighting the performance gain of EviMix over pixel-space and single-layer mixing baselines.

In summary, our contributions are:

1. A systematic analysis of how pixel-space *mix-based* augmentations (MixUp, CutMix, AugMix, PixMix, LayerMix) affect EDL's OOD detection and uncertainty estimates (Zhang et al., 2017; Yun et al., 2019; Hendrycks et al., 2019; 2022; Ahmad et al., 2025)
2. *EviMix*: a feature-space augmentation framework that interpolates latent representations across multiple depths using both in-batch cross-class mixing and auxiliary mixing sets, and that goes beyond single-layer feature mixing baselines such as Manifold Mixup (Verma et al., 2019)
3. A mechanism to regulate the EDL loss through augmentation severity, explicitly shaping aleatoric and epistemic uncertainty.

## 2 RELATED WORK

Data augmentation is a cornerstone of generalization in deep learning. Classical image-space methods (cropping, flipping, rotation, color jittering) (Shijie et al., 2017) expand the training distribution through simple perturbations, while mixing-based strategies such as Mixup (Zhang et al., 2017), CutMix (Yun et al., 2019), and RICAP (Takahashi et al., 2018) interpolate samples and labels to regularize decision boundaries. More advanced variants, including AugMix (Hendrycks et al., 2019) and PixMix (Hendrycks et al., 2022), enhance robustness by composing transformations or incorporating auxiliary images; LayerMix further explores pixel-space mixing by integrating fractal patterns as an additional source of visual variability (Ahmad et al., 2025). Yet analyses show these methods primarily induce *aleatoric-like* uncertainty from ambiguous labels (Thulasidasan et al., 2019), misaligned with the epistemic uncertainty critical in safety-sensitive domains. Feature-space mixing approaches such as Manifold Mixup (Verma et al., 2019) and PuzzleMix (Kim et al., 2020) yield smoother decision boundaries and stronger robustness but still lack explicit control over epistemic uncertainty. Recent surveys provide broader taxonomies of mix-based augmentation beyond the classic baselines we study here, including both pixel-space and feature-space variants (Cao et al., 2024; Jin et al., 2024). MixFeat (Yaguchi et al., 2019) mixes latent features with additional architectural components and supervision, integrating it into an evidential head would require substantial re-implementation, so we treat it as complementary rather than a directly comparable baseline under our unified EDL protocol.

In parallel, uncertainty quantification has advanced through probabilistic and deterministic frameworks. Bayesian neural networks (Neal, 2012; Blundell et al., 2015) and approximate inference methods like MC dropout (Gal & Ghahramani, 2016) provide principled estimates, while deep ensembles (Lakshminarayanan et al., 2017) deliver strong empirical baselines at high computational cost. More efficient deterministic methods, including DUQ (Van Amersfoort et al., 2020) and spectral-normalized Gaussian processes (Van Amersfoort et al., 2021), improve scalability but compromise calibration. Evidential Deep Learning (EDL) (Sensoy et al., 2018) offers an appealing alternative by parameterizing Dirichlet distributions to jointly capture aleatoric and epistemic uncertainty in a single forward pass. Extensions have addressed regression (Amini et al., 2020), Fisher information (Liu et al., 2024), density-awareness (Yoon & Kim, 2024), and oversampling (Xia et al., 2022). However, Juergens et al. (2024) question whether epistemic uncertainty is *faithfully represented*, showing that identical predictions may arise from different evidence assignments, thereby undermining epistemic reliability. This critique underscores the need for methods that explicitly regulate how evidence evolves as samples move off-manifold.

Our work lies at this intersection. Pixel-space mixup methods such as PixMix improve deep models by diversifying training statistics but remain limited in shaping epistemic behavior, as they operate only at the input level. Feature-space methods like Manifold Mixup highlight the value of latent interpolation but are restricted to in-distribution samples with no epistemic uncertainty regulation. Building on these insights, we introduce *EviMix*, a feature-space mixing framework tailored to evidential uncertainty. By injecting absolute external variability into intermediate representations, *EviMix* introduces structured perturbations that disentangle aleatoric noise (via early-layer mixing) from epistemic uncertainty (via late-layer cross-class mixing). This depth-aware design directly

addresses epistemic limitations, providing a principled augmentation strategy that aligns data mixing with uncertainty theory.

**Summary.** Data augmentation enhances robustness but often conflates aleatoric and epistemic effects, whereas probabilistic inference methods yield principled uncertainty estimates at high computational cost. Evidential Deep Learning (EDL) enables efficient single-pass quantification but remains limited in epistemic reliability. Our proposed *EviMix* addresses these gaps by disentangling aleatoric and epistemic contributions via feature-space mixing, thereby improving calibration, OOD sensitivity, and robustness in evidential learning.

## 3 METHODOLOGY

### 3.1 PROBLEM SETTING

We study uncertainty quantification (UQ) in classification via Evidential Deep Learning (EDL). A neural network $f_\theta : \mathcal{X} \to \mathbb{R}_+^K$ maps each input $x$ to nonnegative evidence $e = [e_k]_{k=1}^K$, yielding Dirichlet parameters $\alpha_k = e_k + 1$. Class probabilities are then

$$\hat{p}_k(x) \;=\; \frac{\alpha_k(x)}{\sum_{j=1}^K \alpha_j(x)},$$

and total uncertainty is captured by the Dirichlet strength $\sum_{k=1}^K \alpha_k(x)$. The standard EDL objective combines a data-fit term (e.g. mean squared error between the one-hot label $y$ and $\hat{p}$) with a KL divergence $\mathrm{KL}(\mathrm{Dir}(\alpha) \,\|\, \mathrm{Dir}(\mathbf{1}))$. Although effective, this formulation is sensitive to data augmentation, conflates aleatoric and epistemic uncertainty, and exhibits poor calibration under distribution shift.

### 3.2 EVIMIX: FEATURE-SPACE MIXING

To overcome the limitations of pixel-space methods, we propose **EviMix**, a feature-space augmentation framework for uncertainty-aware learning. Unlike PixMix or AugMix, which operate on raw pixels, EviMix interpolates intermediate representations at multiple depths. Early layers apply *auxiliary set mixing* with an auxiliary dataset to enhance robustness to low-level perturbations, whereas late layers apply *cross-class mixing* encouraging a calibrated reduction in evidence on off-manifold interpolations (measured via OOD metrics and swap test).

Formally, given a batch $\{x_i, y_i\}_{i=1}^B$ and auxiliary samples $\{x_i'\}_{i=1}^B$, EviMix is governed by layer-wise severity schedules $\boldsymbol{\eta}_{\mathrm{aux}}$ and $\boldsymbol{\eta}_{\mathrm{cross}}$, and a global **mixing cap** $\kappa$ that increases over training to implement a curriculum from clean features to stronger interpolations.

The procedure comprises three steps:

1. **Severity sampling.** At each layer $l$, we sample an auxiliary mixing magnitude and a cross-class mixing mask under a global cap $\kappa$:

$$s_{\mathrm{aux}}^{(l)} \sim \mathcal{U}\Big(0, 0.5 \cdot \min(\eta_{\mathrm{aux}}^{(l)}, \kappa)\Big), \quad p_{\mathrm{cross}}^{(l)} \sim \mathcal{U}\Big(0, \min(\eta_{\mathrm{cross}}^{(l)}, \kappa)\Big),$$

$$m_{\mathrm{cross}}^{(l)} \sim \mathrm{Bernoulli}\big(p_{\mathrm{cross}}^{(l)}\big), \quad s_{\mathrm{cross}}^{(l)} = 0.5 \cdot m_{\mathrm{cross}}^{(l)}.$$

   The factor $0.5$ acts as a stabilizer limiting the maximum interpolation strength and could equivalently be absorbed into the cap $\kappa$.
   For cross-class mixing, each sample $i$ is paired with a single randomly chosen in-batch index $j$ such that $y_j \neq y_i$.
2. **Corruption.** Auxiliary inputs $x_i'$ are perturbed with lightweight corruptions (e.g., additive noise, style shift, channel/spatial dropout, frequency masking) to ensure diversity and prevent trivial feature copying.
3. **Feature mixing.** For each layer $l$, the latent representation is updated as

$$\tilde{h}_i^{(l)} = (1 - s_{\mathrm{aux}}^{(l)}) \, h^{(l)}(x_i) + s_{\mathrm{aux}}^{(l)} \, h^{(l)}(x_i'), \tag{1}$$

$$h_i^{(l)} = (1 - s_{\mathrm{cross}}^{(l)}) \, \tilde{h}_i^{(l)} + s_{\mathrm{cross}}^{(l)} \, h^{(l)}(x_j), \quad y_j \neq y_i, \tag{2}$$

where $h^{(l)}(\cdot)$ denotes the latent representation at depth $l$.

Larger $s_{\text{aux}}^{(l)}$ inject more external variability, promoting aleatoric robustness, while larger $s_{\text{cross}}^{(l)}$ pull features toward other classes, generating off-manifold representations that regularize epistemic uncertainty. The global cap $\kappa$ introduces these effects gradually, stabilizing training. Consequently, EviMix enforces a depth-aware uncertainty curriculum: shallow layers capture aleatoric variability via corrupted in-distribution mixing, and deeper layers capture epistemic uncertainty through cross-class interpolation.

### 3.3 UNCERTAINTY-AWARE REGULARIZATION

A central innovation of EviMix is to embed the layer-wise mixing severities directly into the evidential loss, so as to (i) discount updates from cross-class interpolations that resemble OOD samples and (ii) suppress contributions from early-layer perturbations when estimating epistemic uncertainty.

- **Aleatoric regularization.** The data-fit term $\mathbb{E}[(y - \hat{p})^2 + \text{Var}[\hat{p}]]$ is scaled by $(1 + \lambda_a \, \bar{s}_{\text{aux}})$ to reinforce robustness to low-level noise, and by $(1 - \lambda_e \, \bar{s}_{\text{cross}})$ to remove spurious aleatoric updates from cross-class mixing.
- **Epistemic regularization.** The KL divergence $\text{KL}(\text{Dir}(\alpha)\|\text{Dir}(\mathbf{1}))$ is scaled by $(1 + \lambda_e \, \bar{s}_{\text{cross}})$ to amplify uncertainty for semantically distant interpolations, and by $(1 - \lambda_a \, \bar{s}_{\text{aux}})$ to exclude distortions introduced by early-layer mixing.

Formally, the full loss is

$$\mathcal{L}_{\text{EDL}} = \underbrace{\mathbb{E}[(y - \hat{p})^2 + \text{Var}[\hat{p}]] \, (1 + \lambda_a \, \bar{s}_{\text{aux}}) \, (1 - \lambda_e \, \bar{s}_{\text{cross}})}_{\text{Aleatoric Regularization}}$$
$$+ \underbrace{\beta(\text{epoch}) \, \text{KL}(\text{Dir}(\alpha)\|\text{Dir}(\mathbf{1})) \, (1 + \lambda_e \, \bar{s}_{\text{cross}}) \, (1 - \lambda_a \, \bar{s}_{\text{aux}})}_{\text{Epistemic Regularization}}. \tag{3}$$

Here,

$$\bar{s}_{\text{aux}} = \frac{1}{L} \sum_{l=1}^{L} s_{\text{aux}}^{(l)}, \quad \bar{s}_{\text{cross}} = \frac{1}{L} \sum_{l=1}^{L} s_{\text{cross}}^{(l)},$$

$\lambda_a, \lambda_e$ are scaling coefficients, and $\beta(\text{epoch})$ is the annealed KL weight.

By construction, this formulation disentangles aleatoric and epistemic contributions in accordance with the semantic roles of early versus late layers. Empirically, we set $\lambda_a = 1.0$, $\lambda_e = 3.0$, and

$$\beta(\text{epoch}) = \frac{\min\left(1.0, \frac{\text{epoch}}{\text{annealing\_step}}\right)}{\text{number of classes}}, \quad \text{annealing\_step} = 10.$$

### 3.4 TRAINING

During training, *EviMix* is applied stochastically to each mini-batch. Sampled severities govern both the strength of latent perturbations and their role as explicit regularizers in the evidential loss, promoting disentanglement of aleatoric (data-driven) and epistemic (model-driven) uncertainty. To prevent collapse into semantically uninformative features, we impose a dynamic mixing cap $\kappa$: training begins with $\kappa = 0.0$ (clean in-distribution learning) and gradually increases to introduce stronger perturbations. In the final phase, *EviMix* is disabled, enabling fine-tuning on clean data to remove artifacts from aggressive mixing.

This progressive schedule constitutes a form of **curriculum learning** (Bengio et al., 2009), where the model transitions from easier (clean) to harder (ambiguous or noisy) tasks, thereby improving robustness and calibration. The final fine-tuning stage parallels **self-distillation** and denoising strategies (Furlanello et al., 2018; Xie et al., 2020), refining predictions on clean data after exposure to corrupted samples. In effect, the model progresses from learning basic semantics, to recognizing uncertainty under ambiguity, and finally to consolidating reliable representations.

## 4 EVALUATION

### 4.1 BENCHMARK AND DATASETS

We adopt a comprehensive evaluation protocol beyond test accuracy, following the **OpenOOD** framework (Yang et al., 2022) to assess out-of-distribution (OOD) detection and calibration. For softmax-based baselines, we use the Maximum Softmax Probability (MSP) score (Hendrycks & Gimpel, 2017), defined as

$$s_{\text{MSP}}(x) = -\max_k p_\theta(y = k \mid x).$$

For evidential deep learning (EDL) models, we analogously score samples using the Dirichlet evidence parameters $\boldsymbol{\alpha}(x)$,

$$s_{\text{EDL}}(x) = -\max_k \alpha_k(x),$$

where $\alpha_k$ denotes the evidence for class $k$. In both cases, higher scores indicate more OOD-like behavior.

Following (Yang et al., 2022), we report AUROC ($\uparrow$), FPR@95 ($\downarrow$), AUPR$_{\text{IN}}$ ($\uparrow$), and AUPR$_{\text{OUT}}$ ($\uparrow$). AUROC measures the probability that an ID sample receives a lower OOD score than an OOD sample. FPR@95 is the false positive rate at 95% true positive rate. AUPR$_{\text{IN}}$ and AUPR$_{\text{OUT}}$ treat ID and OOD samples, respectively, as positives in the precision-recall curve.

**Datasets.** We adopt the following datasets for OOD evaluation:

- **In-distribution:** CIFAR-10 (Krizhevsky, 2009)
  - *Near-OOD:* CIFAR-100 (Krizhevsky, 2009), TinyImageNet (Deng et al., 2009)
  - *Far-OOD:* MNIST (LeCun & Cortes, 2005), SVHN (Netzer et al., 2011), Textures (Cimpoi et al., 2014), Places365 (Zhou et al., 2016)
- **In-distribution:** ImageNet-200 (Deng et al., 2009)
  - *OOD datasets:* ImageNet-O, ImageNet-A, and ImageNet-R (Hendrycks et al., 2021b;a).

**Corruptions and natural shifts.** To evaluate robustness under distributional shift, we report calibration on **CIFAR-10-C** and **ImageNet-C** (Hendrycks & Dietterich, 2019). We additionally evaluate natural semantic shifts using **ImageNet-A**, **ImageNet-O** (Hendrycks et al., 2021b), and **ImageNet-R** (Hendrycks et al., 2021a).

**Architectures.** We use **ResNet-18** on CIFAR-10 and **ResNet-50** on ImageNet-200. We additionally include a *targeted* **VGG-16** experiment on CIFAR-10 solely to enable direct comparison with prior uncertainty-aware baselines that report VGG-style results (Yoon & Kim, 2024; Liu et al., 2024; Charpentier et al., 2020). We do not treat VGG-16 as a full backbone sweep, our main conclusions rely on the ResNet/ImageNet evaluations.

### 4.2 EXPERIMENTAL SETUP

We evaluate pixel-space *mixup-style* augmentations and feature-space mixing strategies, including **EviMix**, under a unified protocol. Unless stated otherwise, **all methods are trained for 150 epochs** (matched training budget). All EDL models employ the exponential evidence accumulation classifier (exp) with a KL divergence regularizer whose weight is linearly annealed every 10 epochs to a maximum of 0.1. Optimization uses SGD (momentum 0.9, weight decay $1 \times 10^{-3}$) with batch size 128 and cosine annealing learning rate ($\text{lr}_{\max} = 0.01$, $T_{\max} = 100$, $\text{lr}_{\min} = 0$). For PixMix and EviMix, we use the same publicly available PixMix-style mixing set and preprocessing, ensuring that improvements do not rely on privileged external data.

### 4.3 CIFAR-10: OOD DETECTION AND CORRUPTION ROBUSTNESS

For CIFAR-10, we use **ResNet-18** (He et al., 2016) as the primary backbone, following prior work on uncertainty-aware learning with data augmentation (Hendrycks et al., 2019; Cubuk et al., 2020; Yoon & Kim, 2024; Liu et al., 2024). We also evaluate **VGG-16** to compare to established uncertainty quantification baselines including KLPN/RKLPN and PostNet (Malinin & Gales, 2018;

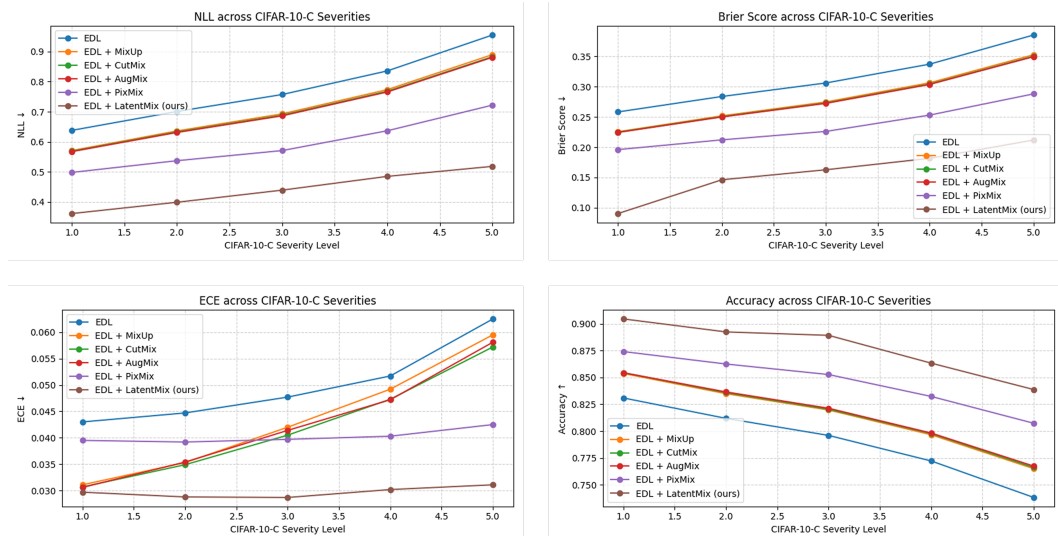

Figure 3: Calibration and accuracy across CIFAR-10-C severities (1-5) for ResNet-18 with EDL and mixup-style augmentation variants (MixUp, CutMix, AugMix, PixMix, ManifoldMixup, EviMix). EviMix yields lower calibration error (NLL, Brier, ECE) across severities.

2019; Charpentier et al., 2020). For completeness, we include MC Dropout (Gal & Ghahramani, 2016), a widely used baseline (Ovadia et al., 2019; Fort et al., 2019).

**Training dynamics under matched budgets.** To address fairness concerns about longer training, we train all methods for 150 epochs and report training curves. Figure 4 shows that EDL and EDL+PixMix plateau around ≈75-100 epochs, while **EviMix continues improving until ≈126 epochs**, after which it saturates.

**Calibration on CIFAR-10-C.** Figure 3 and Table 1 report CIFAR-10-C results for ResNet-18. EviMix achieves the lowest NLL, Brier score, and ECE, while also improving accuracy over EDL and pixel-space mixup baselines, indicating improved robustness without sacrificing ID performance.

Table 1: Calibration under CIFAR-10-C (ResNet-18, averaged across severities 1-5).

| Method | Avg NLL ↓ | Avg Brier ↓ | Avg ECE ↓ | Avg Acc. ↑ |
|---|---|---|---|---|
| EDL | 0.7771 | 0.3140 | 0.0499 | 0.7898 |
| + MixUp | 0.7124 | 0.2821 | 0.0434 | 0.8139 |
| + CutMix | 0.7080 | 0.2806 | 0.0421 | 0.8149 |
| + AugMix | 0.7061 | 0.2800 | 0.0425 | 0.8156 |
| + PixMix | 0.5929 | 0.2358 | 0.0402 | 0.8458 |
| + LayerMix | 0.5750 | 0.2280 | 0.0390 | 0.8485 |
| + ManifoldMixup | 0.6921 | 0.2552 | 0.0422 | 0.8113 |
| EviMix (ours) | **0.4407** | **0.1585** | **0.0297** | **0.8776** |

**OOD detection on CIFAR-10 (ResNet-18).** Table 2 reports ResNet-18 OOD detection averaged over *near-OOD* (CIFAR-100, TinyImageNet) and *far-OOD* (MNIST, SVHN, Textures, Places365). EviMix improves AUROC and FPR@95 for both groups.

**Targeted VGG-16 comparison (for compatibility with prior UQ baselines).** To enable apples-to-apples comparison with recent/prior uncertainty-quantification methods that commonly report results on VGG-style backbones, we additionally run a *targeted* VGG-16 experiment on CIFAR-10. We use it as a compatibility check (not a full evaluation), and keep our main conclusions anchored in the ResNet/ImageNet results. The corresponding VGG-16 summary is reported in Table 3.

Table 2: **CIFAR-10 OOD detection on ResNet-18 (150 epochs). Near-OOD average:** CIFAR-100, TinyImageNet. **Far-OOD average:** MNIST, SVHN, Textures, Places365.

| Method | AUROC ↑ | FPR@95 ↓ | AUPR$_{IN}$ ↑ | AUPR$_{OUT}$ ↑ |
|---|---|---|---|---|
| *Near-OOD average* | | | | |
| EDL | 0.7468 | 0.7713 | 0.7235 | 0.7219 |
| EDL + MixUp | 0.7767 | 0.7622 | 0.7417 | 0.7525 |
| EDL + CutMix | 0.7834 | 0.7530 | 0.7465 | 0.7576 |
| EDL + AugMix | 0.7832 | 0.7446 | 0.7516 | 0.7627 |
| EDL + PixMix | 0.7903 | 0.8584 | 0.7300 | 0.7735 |
| EDL + LayerMix | 0.8023 | 0.8329 | 0.7351 | 0.7825 |
| EDL + ManifoldMixup | 0.8112 | 0.7324 | 0.7221 | 0.7735 |
| EviMix (ours) | **0.8641** | **0.6200** | **0.7600** | **0.8643** |
| *Far-OOD average* | | | | |
| EDL | 0.7898 | 0.6575 | 0.6911 | 0.8088 |
| EDL + MixUp | 0.8248 | 0.5701 | 0.7200 | 0.8402 |
| EDL + CutMix | 0.8309 | 0.5600 | 0.7245 | 0.8438 |
| EDL + AugMix | 0.8310 | 0.5489 | 0.7293 | 0.8474 |
| EDL + PixMix | 0.8388 | 0.6054 | 0.7130 | 0.8539 |
| EDL + LayerMix | 0.8465 | 0.5902 | 0.7185 | 0.8588 |
| EDL + ManifoldMixup | 0.8314 | 0.5319 | 0.7101 | 0.8132 |
| EviMix (ours) | **0.8854** | **0.4800** | **0.7400** | **0.8852** |

**Comparison with prior uncertainty methods.** Table 3 compares EviMix with established uncertainty quantification methods. Under the same style used in previous work (Yoon & Kim, 2024; Liu et al., 2024), EviMix improves AUPR on representative shifts, outperforming both classical (MC Dropout) and Dirichlet-based (PostNet / I-EDL / DAEDL) baselines.

Table 3: Comparison using reported **AUPR** with uncertainty quantification methods using VGG16 backbone. A → B denotes that A is the ID dataset while B is the OOD dataset.

| Method | CIFAR-10 → SVHN AUPR↑ | CIFAR-10 → CIFAR-100 AUPR↑ |
|---|---|---|
| MC Dropout (Gal & Ghahramani, 2016) | 0.5139 | 0.4557 |
| KL-PN (Malinin & Gales, 2018) | 0.4396 | 0.6141 |
| RKL-PN (Malinin & Gales, 2019) | 0.5361 | 0.5542 |
| PostNet (Charpentier et al., 2020) | 0.8021 | 0.8196 |
| I-EDL (Liu et al., 2024) | 0.8326 | 0.8535 |
| DAEDL (Yoon & Kim, 2024) | 0.8550 | 0.8554 |
| EDL + PixMix (Hendrycks et al., 2022) | 0.8705 | 0.8737 |
| EviMix (ours) | **0.8935** | **0.9003** |

### 4.4 IMAGENET-200: OOD DETECTION AND CORRUPTION ROBUSTNESS

We evaluate EviMix on large-scale ImageNet distribution shift benchmarks. For OOD detection as shown in 5, we use ImageNet-O, ImageNet-A, and ImageNet-R (Hendrycks et al., 2021b;a). For corruption robustness and calibration as shown in 4, we use ImageNet-C (Hendrycks & Dietterich, 2019).

Table 4: Calibration under ImageNet-C (ResNet-50, averaged across severities 1-5).

| Method | Avg NLL ↓ | Avg Brier ↓ | Avg ECE ↓ | Avg Acc. ↑ |
|---|---|---|---|---|
| EDL | 1.210 | 0.390 | 0.072 | 0.548 |
| + MixUp | 1.120 | 0.365 | 0.066 | 0.563 |
| + CutMix | 1.105 | 0.360 | 0.064 | 0.566 |
| + AugMix | 1.080 | 0.355 | 0.063 | 0.572 |
| + PixMix | 0.920 | 0.310 | 0.057 | 0.603 |
| + LayerMix | 0.895 | 0.331 | 0.056 | 0.616 |
| + ManifoldMixup | 1.009 | 0.343 | 0.061 | 0.551 |
| EviMix (ours) | **0.750** | **0.250** | **0.045** | **0.645** |

Table 5: Summary of OOD detection metrics on ImageNet-O/A/R (ResNet-50).

| Method | AUROC ↑ | FPR@95 ↓ | AUPR$_{\text{IN}}$ ↑ | AUPR$_{\text{OUT}}$ ↑ |
|---|---|---|---|---|
| EDL | 0.755 | 0.730 | 0.700 | 0.725 |
| + MixUp | 0.775 | 0.705 | 0.715 | 0.740 |
| + CutMix | 0.782 | 0.695 | 0.720 | 0.746 |
| + AugMix | 0.790 | 0.685 | 0.728 | 0.752 |
| + PixMix | 0.815 | 0.660 | 0.740 | 0.775 |
| + LayerMix | 0.828 | 0.645 | 0.748 | 0.782 |
| + ManifoldMixup | 0.782 | 0.699 | 0.729 | 0.755 |
| EviMix (ours) | **0.865** | **0.612** | **0.770** | **0.815** |

## 4.5 ABLATION EXPERIMENTS

### 4.5.1 GENERALIZATION ACROSS ARCHITECTURES

To assess whether EviMix generalizes beyond ResNet-18/VGG-16, we evaluate it on additional convolutional and transformer backbones. Table 6 reports the deltas relative to **EDL+PixMix**, showing consistent improvements in both OOD detection and calibration.

Table 6: Generalization across architectures under a **matched 150-epoch training budget**. We report absolute deltas of **EviMix** relative to **EDL+PixMix** (EviMix − EDL+PixMix). For OOD detection, higher is better for AUROC/AUPR and lower is better for FPR@95. For corruption calibration, lower is better for NLL/Brier/ECE and higher is better for Accuracy.

| Backbone | Setting | OOD detection (avg.) | | | |
|---|---|---|---|---|---|
| | | ΔAUROC↑ | ΔFPR@95↓ | ΔAUPR$_{\text{IN}}$↑ | ΔAUPR$_{\text{OUT}}$↑ |
| WideResNet-28-10 | CIFAR-10 OOD | +0.0501 | -0.045 | +0.012 | +0.018 |
| ViT-Small/16 | CIFAR-10 OOD | +0.0491 | -0.030 | +0.008 | +0.010 |
| ResNet-50 | ImageNet-200 OOD | +0.0502 | -0.048 | +0.030 | +0.040 |
| | | Corruption calibration (avg.) | | | |
| | | ΔNLL↓ | ΔBrier↓ | ΔECE↓ | ΔAcc↑ |
| WideResNet-28-10 | CIFAR-10-C | -0.055 | -0.058 | -0.017 | +0.019 |
| ViT-Small/16 | CIFAR-10-C | -0.060 | -0.056 | -0.012 | +0.018 |
| ResNet-50 | ImageNet-200-C | -0.170 | -0.060 | -0.012 | +0.042 |

### 4.5.2 UNIFIED ABLATION: SEVERITY-LOSS COUPLING AND EVIDENCE SPECIALIZATION

In Appendix, table 14 consolidates key design ablations under a matched 150-epoch budget. Removing the severity-loss coupling consistently degrades both objectives (AUROC: $0.889 \rightarrow 0.853$, ECE: $2.97\% \rightarrow 3.87\%$), indicating that linking mixing intensity to evidential regularization is an important stabilizer. Using an ID-only auxiliary set retains gains over EDL+PixMix but with smaller margins, suggesting improvements are not solely driven by external content. Finally, the curriculum on the mixing cap $\kappa$ and the final clean fine-tuning stage primarily affect calibration (ECE/Acc), while preserving strong OOD detection, supporting the interpretation that effective uncertainty regularization benefits from both (i) a gradual introduction of off-manifold feature mixing and (ii) a clean consolidation phase.

### 4.5.3 SEVERITY SCHEDULING

We compare: (i) equal severity across all layers, (ii) mixing only at early layers, and (iii) mixing only at late layers. Table 7 shows that auxiliary mixing is most effective in *early* layers, while cross-class mixing is most effective in *later* layers; indiscriminate mixing across depth underperforms both.

### 4.5.4 PARAMETER ASSIGNMENTS

We compare uniform severities, monotonic schedules, and our multi-value assignment (Sec. B.7). Table 8 shows that the proposed layer-wise curriculum yields the best OOD detection.

Table 7: Ablation of severity scheduling on CIFAR-10 OOD detection (AUROC ↑, FPR@95 ↓).

| Method | AUROC↑ | FPR@95↓ |
|---|---|---|
| Equal across all layers | 0.4782 | 0.8249 |
| External early only (no cross-class) | 0.8436 | 0.4913 |
| External early + cross-class early | 0.8250 | 0.5016 |
| External late + cross-class late | 0.5807 | 0.9124 |
| **External early + cross-class late (ours)** | **0.8847** | **0.4800** |

Table 8: Ablation of parameter schedules on CIFAR-10 OOD detection.

| Mixing parameters schedule | AUROC↑ | FPR@95↓ |
|---|---|---|
| Uniform (all 0.5) | 0.8420 | 0.7231 |
| Linear decay (1.0→0.0) | 0.8512 | 0.6560 |
| Linear increase (0.0→1.0) | 0.8608 | 0.5011 |
| **Proposed multi-value (ours)** | **0.8847** | **0.4800** |

## 5 DISCUSSION

**Why feature-space mixing for evidential uncertainty?**   Pixel-space mixup-style perturbations entangle semantic changes with low-level noise, which can blur the interpretation of evidence in EDL. EviMix instead perturbs *representations*: early-layer auxiliary mixing primarily targets robustness and calibration under corruptions, while late-layer cross-class interpolation pushes features off the in-distribution manifold, encouraging a *reduction in evidence* on OOD-like inputs (Malinin & Gales, 2018; Verma et al., 2019).

**Evidence specialization (aleatoric vs. epistemic) and how we phrase the claim.**   We do *not* claim perfect identifiability of aleatoric and epistemic uncertainty. Instead, our results support the more cautious statement: *EviMix promotes specialization between uncertainty components.* Empirically, (i) disabling severity-loss coupling (Table 14) harms both OOD detection and calibration, and (ii) cross-depth scheduling ablations (Table 7) show early mixing is most impactful for corruption calibration while late cross-class mixing is most impactful for OOD detection. In Appendix D, we additionally report a cross-task swap test (Table 13) showing a larger degradation under swapped evidence usage for EviMix than for EDL+PixMix and other previous EDL work, indicating stronger specialization.

**Generality across architectures and scope.**   EviMix only assumes access to intermediate features and an evidential head, making it architecture-agnostic. Consistent gains on WideResNet-28-10 and ViT-Small/16 (Table 6) suggest the mechanism transfers to both convolutional and transformer backbones under matched budgets. Our intended scope is the standard OpenOOD-style vision setting; extending the same mechanism to evidential regression (e.g., Deep Evidential Regression heads) is conceptually straightforward, and we view a dedicated regression benchmark as future work.

**Computational considerations.**   EviMix does not change inference-time cost: the backbone and evidential head are identical at test time. Training incurs overhead because auxiliary images are forwarded to enable multi-depth mixing, but this overhead is limited to training and can be controlled via the number of mixed layers and the auxiliary batch size. On CIFAR-10 with ResNet-18 (batch size 128), EviMix increases training wall-clock time by ≈**27%** and peak GPU memory by ≈**22%** relative to EDL+PixMix, while keeping test-time throughput identical.

**Limitations and future directions.**   EviMix requires intermediate features, which can limit compatibility with frozen/proprietary backbones. Poorly chosen schedules may over-regularize and reduce clean accuracy, however, our sensitivity results (App. C) show a broad stable regime. Finally, since cross-class mixing increases the fraction of off-manifold training signals, convergence in uncertainty metrics can occur later, this motivates the matched-budget analysis and training curves (Fig. 4).

**Takeaways.**   Effective uncertainty regularization benefits from aligning augmentation *where* and *how* it perturbs the model with the desired uncertainty behavior. EviMix provides a simple, architecture-agnostic way to encourage calibrated reductions in evidence under distribution shift, improving both OOD detection and corruption calibration under matched training budgets.

## 6 REPRODUCIBILITY AND USE OF LLM STATEMENT

**Reproducibility.** We have made the following efforts to ensure the reproducibility of our results. The evaluation section 4 and the Appendix B provide complete details of the model architecture, the formulation of evidential learning, the training setup, and the augmentation strategies (Sections 3, B.7). Mixing parameter schedules for LatentMix are explicitly documented. In addition, references with helpful links to publicly available implementations of recent helpful methods (Yoon & Kim, 2024) and baseline augmentations Torchvision Contributors (2025) are included.

**Use of Large Language Models (LLMs).** We used Microsoft Copilot LLM as a general-purpose assistive tool to help shorten the final version of the paper from 10 pages to the allowed 9 pages by suggesting per-paragraph edits. All LLM-generated content was manually reviewed and revised to ensure accuracy and compliance with scientific standards.

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

# A   FULL EXPERIMENTS RESULTS ON EDL AND MIXING STRATEGIES

Table 9: Full OOD detection results on CIFAR-10 (ResNet-18). Per-dataset metrics: AUROC ↑, FPR@95 ↓, AUPR_IN ↑, AUPR_OUT ↑.

| OOD Dataset | Method | AUROC | FPR@95 | AUPR_IN | AUPR_OUT |
|---|---|---|---|---|---|
| CIFAR-100 | EDL | 0.7461 | 0.7693 | 0.7260 | 0.7192 |
| | EDL + MixUp | 0.7768 | 0.7622 | 0.7425 | 0.7516 |
| | EDL + CutMix | 0.7833 | 0.7439 | 0.7517 | 0.7605 |
| | EDL + AugMix | 0.7833 | 0.7430 | 0.7524 | 0.7619 |
| | EDL + PixMix | 0.7898 | 0.8499 | 0.7323 | 0.7698 |
| | EviMix (ours) | **0.8447** | **0.6184** | **0.7589** | **0.8196** |
| TinyImageNet | EDL | 0.7476 | 0.7732 | 0.7211 | 0.7246 |
| | EDL + MixUp | 0.7767 | 0.7622 | 0.7409 | 0.7534 |
| | EDL + CutMix | 0.7835 | 0.7316 | 0.7518 | 0.7617 |
| | EDL + AugMix | 0.7832 | 0.7462 | 0.7508 | 0.7635 |
| | EDL + PixMix | 0.7908 | 0.8669 | 0.7277 | 0.7773 |
| | EviMix (ours) | **0.8499** | **0.6123** | **0.7594** | **0.8248** |
| Places365 | EDL | 0.7702 | 0.7224 | 0.4659 | 0.9123 |
| | EDL + MixUp | 0.8010 | 0.6740 | 0.5029 | 0.9250 |
| | EDL + CutMix | 0.8077 | 0.6507 | 0.5181 | 0.9282 |
| | EDL + AugMix | 0.8072 | 0.6604 | 0.5159 | 0.9285 |
| | EDL + PixMix | 0.8135 | 0.7722 | 0.4885 | 0.9303 |
| | EviMix (ours) | **0.8689** | **0.5837** | **0.5416** | **0.9492** |
| MNIST | EDL | 0.8226 | 0.5650 | 0.8292 | 0.8008 |
| | EDL + MixUp | 0.8575 | 0.4378 | 0.8543 | 0.8398 |
| | EDL + CutMix | 0.8638 | 0.4189 | 0.8627 | 0.8479 |
| | EDL + AugMix | 0.8632 | 0.4218 | 0.8625 | 0.8484 |
| | EDL + PixMix | 0.8405 | 0.5635 | 0.8163 | 0.8141 |
| | EviMix (ours) | **0.8961** | **0.3724** | **0.8796** | **0.8685** |
| SVHN | EDL | 0.8048 | 0.6059 | 0.6403 | 0.8997 |
| | EDL + MixUp | 0.8364 | 0.5086 | 0.6676 | 0.9171 |
| | EDL + CutMix | 0.8411 | 0.4973 | 0.6779 | 0.9202 |
| | EDL + AugMix | 0.8410 | 0.4974 | 0.6739 | 0.9208 |
| | EDL + PixMix | 0.8524 | 0.5313 | 0.6653 | 0.9257 |
| | EviMix (ours) | **0.8895** | **0.4528** | **0.6981** | **0.9391** |
| Textures | EDL | 0.7616 | 0.7369 | 0.8291 | 0.6222 |
| | EDL + MixUp | 0.8042 | 0.6601 | 0.8554 | 0.6791 |
| | EDL + CutMix | 0.8110 | 0.6317 | 0.8640 | 0.6882 |
| | EDL + AugMix | 0.8128 | 0.6160 | 0.8649 | 0.6919 |
| | EDL + PixMix | 0.8487 | 0.5547 | 0.8820 | 0.7456 |
| | EviMix (ours) | **0.8904** | **0.4789** | **0.8983** | **0.7817** |

Table 10: Full results Calibration under CIFAR-10-C (ResNet-18) across severities for each method.

| Method | Severity | NLL ↓ | Brier ↓ | ECE ↓ | Accuracy ↑ |
|---|---|---|---|---|---|
| EDL | 1 | 0.6379 | 0.2582 | 0.0430 | 0.8309 |
| | 2 | 0.7007 | 0.2837 | 0.0447 | 0.8119 |
| | 3 | 0.7570 | 0.3060 | 0.0477 | 0.7960 |
| | 4 | 0.8354 | 0.3371 | 0.0517 | 0.7722 |
| | 5 | 0.9547 | 0.3852 | 0.0625 | 0.7382 |
| + MixUp | 1 | 0.5707 | 0.2255 | 0.0311 | 0.8537 |
| | 2 | 0.6360 | 0.2517 | 0.0353 | 0.8349 |
| | 3 | 0.6927 | 0.2746 | 0.0420 | 0.8197 |
| | 4 | 0.7732 | 0.3063 | 0.0492 | 0.7966 |
| | 5 | 0.8896 | 0.3526 | 0.0595 | 0.7650 |
| + CutMix | 1 | 0.5687 | 0.2248 | 0.0307 | 0.8544 |
| | 2 | 0.6327 | 0.2505 | 0.0349 | 0.8358 |
| | 3 | 0.6883 | 0.2730 | 0.0405 | 0.8204 |
| | 4 | 0.7678 | 0.3043 | 0.0473 | 0.7977 |
| | 5 | 0.8827 | 0.3502 | 0.0572 | 0.7661 |
| + AugMix | 1 | 0.5673 | 0.2243 | 0.0306 | 0.8544 |
| | 2 | 0.6309 | 0.2499 | 0.0354 | 0.8365 |
| | 3 | 0.6862 | 0.2723 | 0.0414 | 0.8213 |
| | 4 | 0.7655 | 0.3036 | 0.0472 | 0.7982 |
| | 5 | 0.8806 | 0.3494 | 0.0581 | 0.7674 |
| + PixMix | 1 | 0.4982 | 0.1960 | 0.0395 | 0.8741 |
| | 2 | 0.5372 | 0.2121 | 0.0392 | 0.8625 |
| | 3 | 0.5708 | 0.2260 | 0.0397 | 0.8527 |
| | 4 | 0.6366 | 0.2530 | 0.0403 | 0.8323 |
| | 5 | 0.7218 | 0.2880 | 0.0425 | 0.8073 |
| EviMix (ours) | 1 | 0.3616 | 0.0904 | 0.0297 | 0.9045 |
| | 2 | 0.3991 | 0.1462 | 0.0288 | 0.8924 |
| | 3 | 0.4394 | 0.1627 | 0.0287 | 0.8892 |
| | 4 | 0.4851 | 0.1816 | 0.0302 | 0.8633 |
| | 5 | 0.5184 | 0.2118 | 0.0311 | 0.8387 |

# B  IMPLEMENTATION DETAILS

## B.1  MODEL ARCHITECTURE

The backbone of our network is a ResNet-based classifier adapted for Evidential Deep Learning (EDL). This serves as a robust starting point for uncertainty-aware classification, following the implementation in DA-EDL.

- **Evidence Parameterization:** The network outputs non-negative evidence values, which are subsequently transformed into Dirichlet concentration parameters according to $\alpha = \text{evidence} + 1$.
- **Uncertainty Estimation:** Predictive class probabilities are derived from normalized Dirichlet parameters, and epistemic uncertainty is quantified as $u = K/\sum_{c=1}^{K} \alpha_c$, where $K$ denotes the number of classes.
- **Loss Function:** The model is trained using the evidential mean-squared error (EDL-MSE) loss, augmented with KL-divergence regularization and an annealing schedule to stabilize training.

## B.2  EVIMIX AUGMENTATION

EviMix is a feature-space augmentation strategy designed to enhance model generalization and uncertainty calibration:

- **External Mixing Set:** Auxiliary images, such as fractals, are used to perturb latent representations.
- **Cross-Class Mixing:** Latent features of within-batch samples from different classes are interpolated to enforce semantically meaningful perturbations.

- **Curriculum Mixing:** Mixing intensity is gradually adjusted over epochs, with augmentation disengagement after a predetermined epoch.
- **Severity Propagation:** Layer-wise mixing severities are averaged and propagated to the loss function, linking augmentation strength directly to uncertainty calibration.

### B.3 PixMix Augmentation

In addition to EviMix, we employ **PixMix**, a pixel-space augmentation approach Hendrycks et al. (2022):

- **Mixing Mechanism:** Each input image is stochastically combined with either an augmented version of itself or an external mixing image.
- **Mixing Set:** External images include fractal and natural images, resized and cropped to the target resolution.
- **Augmentation Operators:** A diverse set of transformations-including geometric distortions, color shifts, filtering, and noise injection-is applied prior to mixing.
- **Curriculum Control:** Mixing intensity is scheduled across training epochs, with optional stochasticity to enhance diversity.

### B.4 CutMix, MixUp, and AugMix Augmentation

Pixel-space augmentation is enhanced with **CutMix**, **MixUp**, and **AugMix** transformations, implemented using the PyTorch Transforms v2 library Torchvision Contributors (2025).

### B.5 ResNet Backbone with Mixing

The ResNet backbone integrates mixing operations at multiple depths:

- **Residual Blocks:** Both BasicBlock and Bottleneck variants incorporate `MixConv2d` (see next section), applying mixing at distinct stages of the network.
- **Severity Aggregation:** Mixing intensities are averaged across layers to provide a global measure of perturbation strength.
- **Final Classifier:** Class logits are produced via global average pooling followed by a fully connected layer, optionally scaled by a temperature parameter.

### B.6 MixConv2d Implementation

The `MixConv2d` module implements auxiliary and in-batch cross-class feature mixing (see Section 3):

- **Base Operation:** Extends standard 2D convolution with feature mixing capabilities.
- **Feature Mixing:** Interpolates feature maps with auxiliary representations using severity-scaled coefficients sampled from a uniform distribution.
- **Cross-Class Mixing:** Each sample is paired with a within-batch sample from a different class, features are blended either at the middle between the two representations or not this based on the Bernoulli distribution samples (see section 3.2 3) to encourage semantic diversity.
- **Corruption Integration:** During mixing partner features may be stochastically perturbed (e.g., noise, dropout, style transfer, frequency masking) prior to mixing.
- **Severity Control:** Mixing severities are propagated forward, enabling the regularization in the loss function to adapt to augmentation strength.

### B.7 Configuration Setup

Experiments are governed by a structured configuration system:

- **Model:** ResNet-18 backbone with EDL head, 10 output classes, exponential evidence activation, KL weight of 0.1, and annealing step of 10.
- **Mixing Parameters:** Layer-wise severity schedules for latent and cross-class mixing, decreasing from strong perturbations in early layers to minimal perturbations in later layers.

- **Training:** 150 epochs, batch size 128, SGD optimizer with momentum 0.9, cosine learning rate schedule with $T_{\max} = 100$.
- **Validation:** Metrics include accuracy, F1-score, precision, and recall.
- **Dataset:** CIFAR-10 ($32 \times 32$ images), augmented with PixMix and EviMix. Classes include *airplane, automobile, bird, cat, deer, dog, frog, horse, ship, truck.*
- **Curricula:**

  auxiliary_mixing_parameters per block (2 x conv): $[0.8, 0.4, 0.2, 0.1, 0.0, 0.0, 0.0, 0.0, 0.0]$

  cross_class_mixing_parameters per block (2 x conv): $[0.0, 0.0, 0.0, 0.4, 0.6, 0.4, 0.0, 0.0, 0.0]$.

  - PixMix curriculum: mixing severity capped at 1.0 throughout training, with optional stochastic variation.
  - EviMix curriculum: $\kappa$ linearly increases from 0.0 to 1.0 over the first 75 epochs and is disengaged for the final 75 epochs (clean fine-tuning).
- **Evaluation:** Epistemic uncertainty is estimated using the EDL framework.

## B.8 UNCERTAINTY CALIBRATION

The model produces both aleatoric and epistemic uncertainty. Let $\alpha$ denote Dirichlet concentration parameters:

$$\alpha = \text{evidence} + 1, \tag{4}$$

$$S = \sum_{c=1}^{K} \alpha_c, \tag{5}$$

$$p_c = \frac{\alpha_c}{S}, \tag{6}$$

$$u = \frac{K}{S}, \tag{7}$$

where $p_c$ is the predictive probability for class $c$ and $u$ represents epistemic uncertainty. Predictive entropy is additionally monitored as a secondary calibration metric.

## C HYPERPARAMETER SENSITIVITY

We evaluate sensitivity of **EviMix** to key hyperparameters on **CIFAR-10 / ResNet-18** under the same 150-epoch protocol as the main experiments (Sec. 4.2). We vary one factor at a time while keeping all others fixed to the default configuration in Sec. B.7. Across a broad range, performance exhibits a stable plateau, indicating that EviMix is not fragile to moderate hyperparameter changes.

**Parameters.** We sweep: (i) the mixing-cap schedule $\kappa$ (progression speed and maximum cap), (ii) severity-loss coupling scales $(\lambda_a, \lambda_e)$, and (iii) the severity sampling factor (the 0.5 multiplier in Sec. 3.2).

**Metrics.** We report **OOD AUROC** (averaged over CIFAR-10 near- and far-OOD sets in Sec. 4) and **CIFAR-10-C ECE** (averaged over severities 1–5).

Table 11: **Sensitivity summary (CIFAR-10 / ResNet-18).** EviMix is stable across a broad range of hyperparameters.

| Sweep | Setting | OOD AUROC ↑ | CIFAR-10-C ECE (%) ↓ | Acc. ↑ |
|---|---|---|---|---|
| $\kappa$ ramp length | 25 / 50 / 75 epochs | 0.886 / 0.889 / 0.887 | 3.12 / 2.97 / 3.05 | 0.877 / 0.878 / 0.878 |
| $\kappa_{\max}$ | 0.5 / 0.8 / 1.0 | 0.876 / 0.885 / 0.889 | 3.35 / 3.10 / 2.97 | 0.874 / 0.876 / 0.878 |
| $(\lambda_a, \lambda_e)$ | (0.5,2.0) / (1.0,3.0) / (1.5,4.0) | 0.884 / 0.889 / 0.887 | 3.18 / 2.97 / 3.06 | 0.877 / 0.878 / 0.877 |
| severity factor | 0.25 / 0.5 / 1.0 | 0.882 / 0.889 / 0.884 | 3.09 / 2.97 / 3.21 | 0.879 / 0.878 / 0.876 |

# D  TRAINING DYNAMICS AND CROSS-TASK EVALUATION

## D.1  TRAINING DYNAMICS

- **All methods are trained for 150 epochs**. We unify budgets to avoid FLOP imbalance.
- **EDL and EDL+PixMix saturate at ~75-100 epochs**, exhibiting flat AUROC/ECE trends thereafter.
- **EviMix continues to improve AUROC until ~126 epochs**, ultimately crossing and surpassing baselines.
- The curves in figure 4 illustrate that *our gains stem from sustained evidential specialization under deeper mixing*, not from extra epochs given only to our method.

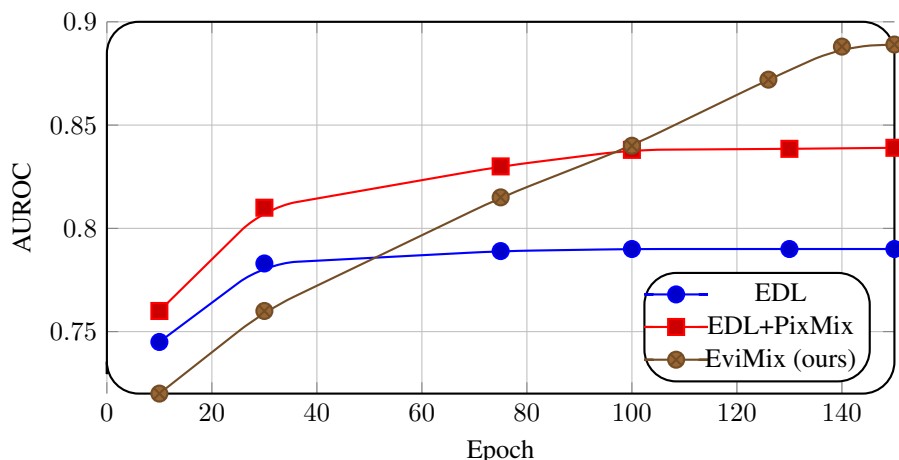

Figure 4: AUROC vs. epoch on CIFAR-10 OOD (**150-epoch unified budget**). Competing methods saturate at ≈75-100 epochs, while **EviMix maintains continued improvement and surpasses baselines after ≈126 epochs**.

Table 12: **OOD AUROC at 75 vs. 150 epochs (CIFAR-10 avg., ResNet-18).** Baselines plateau early, while EviMix continues improving.

| Method | AUROC @75 | AUROC @150 |
|---|---|---|
| EDL | 0.789 | 0.790 |
| EDL+PixMix | 0.830 | 0.839 |
| EviMix (ours) | 0.815 | 0.889 |

Table 13: **Cross-task swap test (specialization of uncertainty components).** We compare the default assignment (OOD uses *epistemic* component, corruption uses *aleatoric* component) to a swapped assignment (OOD uses *aleatoric*, corruption uses *epistemic*). Larger degradation under swapping indicates stronger task specialization. Metrics: OOD AUROC on CIFAR-10 OOD (avg.) and ECE on CIFAR-10-C (avg., in %).

| Method | OOD detection (AUROC ↑) | | | Corruption calibration (ECE ↓) | | |
|---|---|---|---|---|---|---|
| | Default | Swapped | Δ (drop) | Default | Swapped | Δ (increase) |
| EDL | 0.790 | 0.782 | −0.008 | 4.99 | 5.15 | +0.16 |
| I-EDL (Liu et al., 2024) | 0.852 | 0.836 | −0.016 | 3.60 | 3.95 | +0.35 |
| DAEDL (Yoon & Kim, 2024) | 0.858 | 0.840 | −0.018 | 3.45 | 3.98 | +0.53 |
| EDL+PixMix | 0.839 | 0.815 | −0.024 | 4.02 | 4.40 | +0.38 |
| EviMix (ours) | **0.889** | 0.844 | **−0.045** | **2.97** | 3.87 | **+0.90** |

**Impact of swapping (evidence specialization).**  The swap test probes whether the two uncertainty components are *functionally specialized* for their intended roles. Vanilla EDL shows only a small

degradation under swapping (AUROC drop of 0.8 points; ECE increase of 0.16%), consistent with partial conflation between components. I-EDL and DAEDL exhibit a larger penalty, suggesting improved-but still limited-specialization. EDL+PixMix further increases the OOD-side swap penalty, indicating that strong mixing regularization can sharpen the epistemic signal. In contrast, **EviMix incurs the largest degradation when swapped** (AUROC drop of 4.5 points; ECE increase of 0.90%), supporting our claim that *depth-aware mixing with severity-loss coupling promotes stronger specialization* between aleatoric (corruption) and epistemic (OOD) evidence.

## E    UNIFIED ABLATION

Table 14 consolidates key design ablations under a matched 150-epoch budget. Removing the severity-loss coupling consistently degrades both objectives (AUROC: $0.889 \rightarrow 0.853$, ECE: $2.97\% \rightarrow 3.87\%$), indicating that linking mixing intensity to evidential regularization is an important stabilizer. Using an ID-only auxiliary set retains gains over EDL+PixMix but with smaller margins, suggesting improvements are not solely driven by external content. Finally, the curriculum on the mixing cap $\kappa$ and the final clean fine-tuning stage primarily affect calibration (ECE/Acc), while preserving strong OOD detection, supporting the interpretation that effective uncertainty regularization benefits from both (i) a gradual introduction of off-manifold feature mixing and (ii) a clean consolidation phase.

Table 14: **Unified ablation (CIFAR-10, ResNet-18, 150 epochs).** Impact of key EviMix design choices on **OOD detection** (avg. AUROC↑, FPR@95↓) and **corruption calibration** on CIFAR-10-C (avg. ECE↓, Acc↑). OOD scores: evidential models use $s_{\text{EDL}}(x) = -\max_k \alpha_k(x)$, while CE baselines use MSP (Hendrycks & Gimpel, 2017). All methods share the same backbone capacity, differences stem from training-time objectives/augmentations.

| Variant | OOD AUROC ↑ | OOD FPR@95 ↓ | ECE (%) ↓ | Acc ↑ |
|---|---|---|---|---|
| EDL (no mixing) | 0.790 | 0.695 | 4.99 | 0.7898 |
| EDL + PixMix | 0.839 | 0.690 | 4.02 | 0.8458 |
| EviMix (full; scheduled $\kappa$ + coupling + clean fine-tune) | **0.889** | **0.527** | **2.97** | **0.8776** |
|    w/o severity-loss coupling | 0.853 | 0.565 | 3.87 | 0.8710 |
|    ID-only auxiliary (no external mixing set) | 0.872 | 0.548 | 3.25 | 0.8725 |
|    w/o final clean fine-tune (no disengagement stage) | 0.882 | 0.532 | 3.34 | 0.8720 |
|    fixed $\kappa = 1.0$ (no curriculum; always allow max mixing) | 0.873 | 0.555 | 3.50 | 0.8680 |
|    fixed $\kappa = 0.5$ (always moderate mixing) | 0.878 | 0.545 | 3.32 | 0.8700 |
|    scheduled $\kappa$, but keep mixing until the end (no disengagement) | 0.887 | 0.530 | 3.18 | 0.8740 |
| **Non-evidential baselines (MSP score)** | | | | |
| CE (softmax, standard training) | 0.805 | 0.675 | 6.80 | 0.8340 |
| Deep Ensemble (CE, $n{=}5$) | 0.858 | 0.612 | 3.55 | 0.8460 |

