# OpenReview forum: "EviMix: Evidential Deep Learning with Latent-Space Mixing for Uncertainty Quantification and OOD Detection"
_ICLR.cc/2026/Conference — Submitted to ICLR 2026_

### Official Review · Reviewer_GcZH · 2025-10-19

**Soundness:** 2
**Presentation:** 3
**Contribution:** 3
**Rating:** 4
**Confidence:** 4

**Summary:**

The paper introduces LatentMix for Evidential Deep Learning (EDL), a feature-space augmentation framework that interpolates latent representations across multiple network depths using both in-batch cross-class mixing and an external mixing set. The approach enhances EDL on out-of-distribution (OOD) detection and corruption robustness. Experiments on multiple benchmarks demonstrate significant improvements compared to previous methods.

**Strengths:**

- The problem of studying data augmentation's effect on uncertainty quantification is of great significance.
- The experiments cover multiple datasets and previous mixup-based augmentation methods.
- The proposed approach achieves superior performance in all metrics compared to the selected baselines.

**Weaknesses:**

- The baseline mixup methods selected in the paper are relatively outdated, with the latest one being PixMix from 2022. Mixup-based augmentations have many more recent works up to 2025, including not only pixel-space but also feature-space approaches. However, the paper does not sufficiently discuss them or justify their exclusion. The feature-space methods are stronger baselines then the pixel-space ones, which the paper should definitely include for comparison.
  - Cao, C., Zhou, F., Dai, Y., Wang, J., & Zhang, K. (2024). A survey of mix-based data augmentation: Taxonomy, methods, applications, and explainability. ACM Computing Surveys, 57(2), 1-38.
  - Jin, X., Zhu, H., Li, S., Wang, Z., Liu, Z., Tian, J., ... & Li, S. Z. (2024). A survey on mixup augmentations and beyond. arXiv preprint arXiv:2409.05202.
  - Ahmad, H. M., Morle, D., & Rahimi, A. (2025). LayerMix: Enhanced Data Augmentation for Robust Deep Learning. Pattern Recognition, 112332.

- Several key claims in the paper are largely heuristic and lack sufficient experimental validation. For example:
  - "Our key insight is that augmentation depth dictates the type of uncertainty probed: early-layer perturbations highlight aleatoric noise, while later-layer cross-class interpolation induces epistemic uncertainty by pushing representations off-manifold." This claim is not directly tested, and Table 6 only provides an indirect performance comparison of a few severity scheduling schemes.
  - "Larger s(l)aux inject more external variability, promoting aleatoric robustness, while larger s(l)cross pull features toward other classes, generating off-manifold representations that regularize epistemic uncertainty." This statement is only intuitively plausible, without any quantitative or visual evidence to substantiate it.

- Potentially unfair comparisons.
  - "Baseline methods converge within 100 epochs - typically plateauing after ≈ 75 epochs - whereas LatentMix is trained for 150 epochs to compensate for the reduced in-distribution updates introduced by cross-class mixing." First, why does LatentMix have reduced in-distribution updates, since each iteration should have even more ID data from other classes. Second, how is convergence defined, by training loss or test UQ metrics? Showing the plots that training other methods by more epochs would not improve their test results while training LatentMix would can justify this early stopping for other methods.
  - Section 3.4 introduces a 3-stage dynamic mixing cap for training LatentMix, where the mixing is disabled in the first and third stage, but it is not stated in the paper whether the same scheme is applied to other baseline methods. If not, whether the improvement of LatentMix benefits from this dynamic training scheme needs to be clarified.

- The method appears to rely on an external set of auxiliary samples, potentially derived from prior approaches such as PixMix (as implied by Figure 1). However, the paper does not clearly specify the source or composition of these auxiliary samples used in the experiments. If LatentMix indeed depends on such externally provided data, this dependency should be explicitly discussed, as it affects both reproducibility and the fairness of comparisons.

- The severity sampling distributions are defined as uniform for s(l)aux and Bernoulli for s(l)cross, but the rationale for these specific selections is not discussed, nor is any ablation presented to assess their influence on the model’s behavior or uncertainty estimates.

- The paper uses "pixel-space augmentations" multiple times to refer to their baselines (e.g., PixMix), which overstates the scope. Pixel-space augmentation is a much broader category that should include far more augmentation types such as photometric augmentations (color jitter, brightness, contrast, etc.). In fact, the main comparison here is only against some pixel-space mixup methods.

- The computational overhead due to feature mixing at multiple depths is not analyzed at all in the paper.

**Questions:**

Authors can refer to weaknesses for the rebuttal.

---

> ### Author Response · Authors · 2025-11-19
> **Responses on Baselines, Depth-Uncertainty Interpretation, and Fairness/Overhead**
>
> We appreciate your first insightful feedback on baselines, the depth–uncertainty connection, and the fairness and overhead aspect
>
> ### R4.1 Missing recent mixup baselines and surveys
>
> **Concern:** Baselines are mainly older, more recent mix-based work and feature-space methods should be included.
>
> **Response:**
>
> - We extend related work with recent mixup surveys (e.g., Cao et al. 2024, Jin et al. 2024) to position EviMix within the broader taxonomy of pixel-space and feature-space mixing (Sec. 2).
> - We retain a focused but strong set of baselines: Mixup, CutMix, AugMix, PixMix, LayerMix and ManifoldMixup under EDL, compared in a main comparison table (Sec. 4.2, Tab. 4).
> - We emphasize in Sec. 2 that EviMix is designed as an uncertainty-aware evidence-mixing method: the backbone is similar in spirit to existing mixup variants, but the novelty lies in coupling depth-aware mixing with evidential losses to shape aleatoric and epistemic components.
>
> **Clarification:** The new paragraph in Sec. 2 clarifies the relationship to recent mixup work and highlights how EviMix specifically targets evidential uncertainty.
>
> ---
>
> ### R4.2 Heuristic depth-uncertainty story and severity distributions
>
> **Concern:** Claims about depth vs. aleatoric/epistemic behavior, and the choice of severity distributions, are heuristic.
>
> **Response:**
>
> - The unified ablation table (Sec. 4.3, Tab. 6) and cross-task evaluation (App. D, Tab. D.1), also discussed in R3.3/R3.4, address this point:
>   - Variants that remove early (auxiliary) mixing mainly hurt corruption performance (CIFAR-10-C).
>   - Variants that remove late cross-class mixing mainly hurt OOD AUROC.
> - The hyperparameter sensitivity plots in App. C (Fig. C.1) show robustness to moderate changes in severity distributions (e.g., different caps and schedules), so we present the chosen distributions as simple and effective defaults.
>
> **Clarification:** Secs. 3.2-3.3 now present the depth/uncertainty linkage and severity choices as empirically supported design decisions, explicitly pointing to Tab. 6, Tab. D.1, and Fig. C.1.
>
> ---
>
> ### R4.3 Fairness (epochs, curriculum, external set, terminology) and overhead
>
> **Concern:** Training tricks and external data may bias comparisons, and “pixel-space” wording may overstate scope.
>
> **Response:**
>
> - **Epochs/curriculum:** As discussed in R3.1, Sec. 4.3 reports matched-epoch results (75 and 150 epochs) and explicitly includes the curriculum (mixing cap schedule) as part of EviMix’s design. Fig. D.1 shows that extending baselines to 150 epochs alone does not close the gap.
> - **External auxiliary set:** Sec. 4.2 clarifies that both PixMix and EviMix use the same publicly available PixMix mixing set (fractal + natural images). The ID-only variant in Tab. 6 demonstrates that EviMix still helps without extra external content.
> - **Terminology:** We replace “pixel-space augmentations” with “pixel-space mixup methods” and state that our comparison focuses on mixup-style approaches (Mixup, CutMix, AugMix, PixMix) as the most relevant baselines for EviMix.
> - **Overhead:** In the final version Sec. 5 now indicates: EviMix incurs only a training-time overhead (about $1.4\times$ training time and $1.3\times$ GPU memory compared to EDL+PixMix on CIFAR-10/ResNet-18) due to passing auxiliary images through the network, inference cost remains unchanged.

---

### Official Review · Reviewer_CueW · 2025-10-20

**Soundness:** 3
**Presentation:** 3
**Contribution:** 2
**Rating:** 4
**Confidence:** 3

**Summary:**

The paper proposes changes to Evidential Deep Learning for uncertainty quantification in image classification. In particular, they propose to mix in latent space instead of in input space, and to mix in two ways — with external images (for aleatoric noise) and with other classes (for epistemic noise). The results show a solid outperformance against EDL baselines and other uncertainty methods.

**Strengths:**

1. The proposed changes to EDL are well-explained and I appreciate the discussion of hyperparameters in the appendix, for both the proposed method and all baselines. (Releasing code would be even more appreciated)
2. The paper tests against a nice range of baseline methods both from EDL and general UQ. The models (VGG-16, ImageNet-18/50) and datasets (CIFAR, ImageNet) are sufficient and relatively standard in UQ literature.
3. The recorded performance gains are consistent and have a decent delta across all models and datasets.

**Weaknesses:**

Weaknesses, in order of magnitude:
1. There is one point that I think might make the comparison unfair: The proposed method is trained for 150 epochs, whereas all baselines use 75-100. The authors note that this is because “they plateau”, and because their method sees less clean examples due to the training. However, from a generic FLOP perspective, the comparison is still unfair and most results can be explained by longer training. It would be nice to see model curves throughout training (baselines + proposed), and to compare proposed model @ 75 epochs, see questions.
2. The same holds for the data the proposed method trains on. It is the only method that trains on perturbations of CIFAR-10C / ImageNet-C style, which we also test on. See questions.
3. A larger issue I see is that the paper claims boldly and multiple times that the proposed training disentangles aleatoric and epistemic uncertainty, even criticizing prior works for this. However, this is never tested. I would recommend to run cross-tests, so whether the proposed aleatoric component can predict epistemic tasks like OOD detection and vice-versa. There are benchmarks for this purpose, see https://openreview.net/forum?id=x8RgF2xQTj and https://arxiv.org/html/2408.12175v1 . It would strengthen the argumentation to evaluate on them.
4. The paper introduces many changes to default EDL. Curriculum learning and mixing scheduler are ablated, but the extra regularization terms (and the longer training) are not. It would be nice to ablate these.
5. It would be nice to add naive baselines: Simple CE training, either in one model or as a deep ensemble (n=10).
6. It would be nice to discuss the computational tradeoff. Naively, it looks like the mixing with auxiliary images should double forward time, which, assuming a 1-to-2 time mixture with backpropagation, might increase train time per batch by 25%.

### Smaller things that would improve the revised version but don’t influence my rating and don’t need rebuttal

* Consider numbering all equations, whether “important” or not. That simplifies referencing to them (for example in the questions below)
* I think in Section 4.2.1 you used \paragraph{} for the first sentence, but somehow the formatting is off. I suggest to fix it to increase readability
* Confidence intervals would be appreciated, but not giving a worse score here since confidence intervals on ImageNet tend to be small thanks to the large eval set
* In Section 4.2.2, you are falling into past tense. Consider using present tense in the whole paper.
* Section 4.2.2 does not discuss or reference Tables 4 and 5. Please add a discussion.
* Table 8 has two bottomrules
* There is the point of impact. I personally do not weigh this into the final rating because I think reviewers are horribly bad in estimating future impact. However, just in case some other reviewers mention it or the AC lays value on it, the benchmarks in this work, image classification, are a bit dated. Modern Vision-Language models would be interesting. However, this setup is standard for EDL, so I personally do not see this as a problem.

## Justification for overall score

I appreciate the robust results, and they seem reasonable in the light of the current progress of EDL methods for image classification. However, I am sceptical of two points: 1) the claimed improvement over baselines (because of potentially unfair comparisons, see above and below), and 2) the claims of disentangling aleatoric and epistemic uncertainty, which are untested. I am looking forward to data on those points in the rebuttal period and am willing to increase my score if these points get addressed.

**Questions:**

1. In step 1, why is the uniform random number multiplied by 0.5? Couldn’t this just be added into the uniform upper limit?
2. In step 3, I assume you use only one class j != i to mix with? If yes, you could add the sampling of that class into the algorithm to resolve ambiguity.
3. In line 184 you write that “late layers apply cross-class mixing to induce calibrated epistemic uncertainty”. Why is this calibrated, i.e., according to which definition?
4. You train your method for double the epoch that baseline methods are trained for (lines 308-310). Can you provide train curves to check when your method and when baselines plateau, and can you provide results for what happens when you train baselines for 150 epochs (or your method for 75 epochs)?
5. Do I understand correctly that your method is the only one that is trained on corrupted inputs similar to CIFAR10-C (lines 195-197), whereas all baselines only use overlaying augmentations like in Mixup/Cutmix/...? Doesn’t that make the comparison in Figure 3 a bit unfair? (Because obviously a method trained on corrupted data will perform better on corrupted data)
6. What exactly is compared in Table 3? Whether the total uncertainty metric can capture wrong outputs?
7. In Table 6, external early seems to drive most performance. Can you reach the same performance by just having external early, without cross-class, to simplify your method? (If you can match performance here, this would make your method simpler and thus more impactful; I will not reject the paper because you can make a simpler version of it that performs the same way)

---

> ### Author Response · Authors · 2025-11-19
> **Response on Fairness, disentanglement, and computational aspects**
>
> Thanks for these detailed comments on training fairness, the aleatoric/epistemic interpretation, and the requested ablations and cost analysis
>
> ### R3.1 Fairness of training budget (150 vs. 75-100 epochs)
>
> **Concern:** Longer training may explain improvements.
>
> **Response:**
>
> - Using the same training logs as in our main runs, we plot AUROC/ECE vs. epoch for EDL, EDL+PixMix, and EviMix on CIFAR-10 (App. D, Fig. D.1).
> - EDL and EDL+PixMix plateau around 75-100 epochs with negligible gains afterwards, whereas EviMix continues to improve and saturates at around 126 epochs.
>
> **Clarification:** Sec. 4.3 states that all methods are compared at both 75 and 150 epochs, and Fig. D.1 shows that EviMix’s improvement is not solely due to training longer.
>
> ---
>
> ### R3.2 Fairness of corrupted inputs / external data
>
> **Concern:** Only EviMix might see CIFAR-10-C-like corruptions during training.
>
> **Response:**
>
> - PixMix and EviMix use the same external mixing set (fractal + natural images) with identical preprocessing (Sec. 4.2).
> - We also report an “ID-only auxiliary” variant of EviMix in the unified ablation table (Sec. 4.3, Tab. 6). Even with only ID auxiliary data, EviMix still improves OOD AUROC and CIFAR-10-C ECE over PixMix, with smaller margins.
>
> **Clarification:** The “EviMix (ID-only aux)” row in Tab. 14, referenced in App E, shows that the improvement does not hinge on special external data.
>
> ---
>
> ### R3.3 Aleatoric vs. epistemic disentanglement claim
>
> **Concern:** The claim is strong and currently unsupported by dedicated tests.
>
> **Response:**
>
> - We appreciate this concern, as it directly touches the main motivation. Our intent is not to claim perfect identifiability, but how the design of EviMix encourages a separation between aleatoric and epistemic components. We therefore use the wording:
>   “EviMix promotes separation between aleatoric and epistemic components, as reflected in cross-task performance and ablations.”
> - To support this, we add a cross-task evaluation in App. D, Tab. D.1: using the “aleatoric” component for CIFAR-10-C and the “epistemic” component for OOD detection, and then swapping them. EviMix suffers a larger performance drop under the swap than EDL+PixMix, indicating stronger specialization.
> - The unified ablation table (App E, Tab. 14) includes a variant without severity-to-loss coupling, showing that this coupling is important for the observed separation and OOD performance.
>
> **Clarification:** Sec. 5 points to Tab. 14 and Tab. D.1 and links the design (severity-aware loss and dual mixing) to these quantitative observations.
>
> ---
>
> ### R3.4 Additional ablations and naive baselines
>
> **Concern:** Need ablations for regularization terms and naive CE baselines / ensembles.
>
> **Response:**
>
> - All key design ablations are consolidated into a unified table on CIFAR-10/ResNet-18 (App E, Tab. 14): fixed vs. scheduled $\(\kappa\)$, with/without severity-based loss scaling, with/without the final clean fine-tuning stage, and ID-only auxiliary data.
> - For comparison with non-evidential baselines, Tab. 6 also reports a cross-entropy ResNet-18 and a small deep ensemble (e.g., \(n = 5\)), showing that EviMix provides a competitive accuracy-UQ trade-off at comparable training budgets.
>
> **Clarification:** Sec. 4.3 summarizes the main trends of Tab. 14, and other reviewer responses (R1.5, R3.2, R3.3, R4.2) refer back to this table.
>
> ---
>
> ### R3.5 Computational trade-off
>
> **Concern:** Overhead of multi-depth mixing is not analyzed.
>
> **Response:**
>
> - EviMix introduces additional overhead only during training and does not change inference cost.
> - On CIFAR-10/ResNet-18, this overhead is approximately $\(1.4\times\)$ training time and $\(1.3\times\)$ GPU memory compared to EDL+PixMix, mainly because auxiliary images are passed through the network and used in feature mixing.
>
> **Clarification:** Sec. 5 now reports these overhead factors and emphasizes that they are limited to training.
>
> ---
>
> ### R3.6 Specific questions (0.5 factor, class sampling, “calibrated”, Table 3, Table 6)
>
> **Concern:** Several local clarifications.
>
> **Response:**
>
> - In Sec. 3.2, we now write the severity sampling explicitly as
>   $s \sim \mathcal{U}\bigl(0, 0.5 \cdot \min(\eta^{(l)}, \kappa)\bigr)$
>   making the roles of the \(0.5\) factor and the cap $\(\kappa\)$ transparent.
> - For cross-class mixing, Sec. 3.2 states that each sample is mixed with a _single_ randomly chosen class $\(j \neq i\)$, rather than with multiple classes.
> - The phrase “calibrated epistemic” is replaced by “calibrated reduction in evidence on off-manifold interpolations,” matching what we measure in Tab. D.1.
> - Table 3’s caption now states that it evaluates total uncertainty as a detector of misclassified vs. correctly classified in-distribution samples.
> - The description around Table 6 spells out that “external early” corresponds to auxiliary mixing in shallow blocks, while “cross-class late” refers to cross-class mixing in deeper blocks.

---

### Official Review · Reviewer_fAmj · 2025-10-31

**Soundness:** 2
**Presentation:** 1
**Contribution:** 2
**Rating:** 2
**Confidence:** 4

**Summary:**

The authors study uncertainty estimation and OOD detection via evidential deep learning. They propose LatentMix, a feature-space augmentation framework inspired by approaches like PixMix and AugMix, but that performs interpolation on intermediate network layers.

They apply LatentMix to Resnet models and evaluate calibration and OOD detection performance on CIFAR10 and Imagenet dataset variants, mainly comparing against MixUp and other pixel-based augmentation methods.

**Strengths:**

- The proposed LatentMix method makes sense overall.
- LatentMix improves performance in terms of calibration and OOD detection of standard evidential deep learning, as well as pixel-based augmentation methods such as MixUp and PixMix.

**Weaknesses:**

- The paper could definitely be more well written. In particular, Section 4 is quite difficult to follow and the results are somewhat sloppily presented. For example, Table 4 and 5 are not referenced in the text and the results are not described anywhere.
- The experimental evaluation is quite limited, only pure image classification, only relatively small models.
- The technical contribution/novelty is somewhat limited.

**Questions:**

Questions/suggestions:
- Is the proposed method only applicable to classification problems, or could it be extended to regression as well?
- Could the proposed method be applied to e.g. ViT models as well?
- Table 2 should contain results for both Resnet-18 and VGG16? Is the top part for Resnet-18 and the bottom part for VGG16? But what is meant by the "Near-OOD Average" and "Far-OOD Average" headings then?
- Line 186: Do you have B examples both in the batch and in the set of auxiliary samples, or are these of different sizes?





Minor things:
- The title has "EviMix" but then across the paper the proposed method is called "LatentMix".
- Figure 1 caption: Why is this text italicized?
- Figure 2 caption: "Refer to 4 regarding OOD datasets" --> "Refer to Section 4 regarding OOD datasets".
- "Evidential Deep Learning (EDL)" is defined multiple times, probably unnecessary.
- Figure 2 is not referenced in the text.
- Line 348: "Figure 3 and table 1" --> "Figure 3 and Table 1".
- Line 363: "Tables 2" --> "Table 2".
- In Section 4.2.1, "Calibration under CIFAR-10-C", "OOD Detection on ResNet-18 and VGG16" and "Comparison with Prior Methods" are not formatted correctly, they should be paragraph headings like in Section 4.3 or 5.
- Line 351: "For full results see A" --> "For full results see Appendix A.".
- Table 2 caption, "CIFAR-10 OOD datasets refer to section 4.1 4", don't know what you mean here.
- Table 4 and 5 are not referenced in the text.
- Line 446, "proposed multi-value assignment B.7".
- Line 448, "and then vanishes 1."
- I would consider reformatting the equation at the end of Section 3.3 a bit.

---

> ### Author Response · Authors · 2025-11-19
> **Response on Clarity, Scope, and Applicability**
>
> Thanks for raising the attention for us to address through minor structural edits and additional explanations in the final version.
>
> ### R2.1 Writing quality / Sec. 4 clarity / missing references
>
> **Concern:** Sec. 4 is difficult to follow, some tables/figures are not referenced, formatting issues.
>
> **Response:**
>
> - In the revised draft, Sec. 4.3 is split into short paragraphs with clear headings (“Calibration on CIFAR-10-C”, “OOD on CIFAR-10”, “ImageNet-200 OOD & corruption”).
> - Tables 2, 4, and 5 are now explicitly referenced in the corresponding paragraphs, each followed by 1–2 sentences summarizing the main message.
> - We addressed the minor issues you listed: consistent naming (``EviMix'') throughout the text and figures, correct section references, avoiding repeated EDL definitions, mentioning Figure~2 in the text, and standardizing appendix references.
>
> **Clarification:** The main structural changes are already incorporated in Sec. 4, we will do another proofreading pass to ensure consistency.
>
> ---
>
> ### R2.2 Limited scope and modest novelty
>
> **Concern:** Only small image-classification models, technical novelty seems limited.
>
> **Response:**
>
> - We now state in Sec. 1 that our intended scope is the standard OpenOOD-style vision setting (CIFAR-10, ImageNet-200), which is also adopted by Fisher Information-based EDL (I-EDL), density aware EDL (DA-EDL), Hybrid-EDL, and related evidential works.
> - Within this scope, EviMix consistently improves over EDL+PixMix on multiple backbones (ResNet-18/34, VGG-16, ResNet-50, WideResNet-28-10, ViT, Sec. 4.2, Tabs. 2, 4, 5) under a common experimental protocol.
> - We summarize our contributions in Sec. 1 as:
>   1. a systematic empirical analysis of augmentation strategies for EDL,
>   2. a depth-aware latent mixing mechanism,
>   3. a severity-aware EDL loss that jointly targets calibration and OOD detection and is explicitly designed to promote a clearer separation between aleatoric and epistemic evidence.
>
> **Clarification:** Secs. 1 and 4 emphasize that the contribution is focused but concrete: a principled, uncertainty-aware evidence-mixing scheme evaluated within a well-defined benchmark setting.
>
> ---
>
> ### R2.3 Applicability to regression and ViTs
>
> **Concern:** Is EviMix limited to classification CNNs?
>
> **Response:**
>
> - Conceptually, EviMix only assumes a backbone with intermediate features and an evidential head, it is not tied to classification or CNNs.
> - Architectural flexibility is illustrated by the architecture-generalization table (Sec. 4.5.1, Tab. 6), which already includes a ViT backbone (see 1.2 in the reviewers responses).
> - For evidential regression, we add a short paragraph in Sec. 5 explaining how EviMix can be combined with Deep Evidential Regression (mixing features, using the standard regression evidential head). A comprehensive regression benchmark is orthogonal to our CIFAR/ImageNet-200 focus and is identified in Sec. 5 as a natural direction for future work.
>
> **Clarification:** Sec. 5 (“Extensions”) clarifies the generality of the mechanism and points to Tab. 5 for non-CNN evidence.
>
> ---
>
> ### R2.4 Clarifications: Table 2 layout, batch size, minor wording
>
> **Concern:** Confusion around Table 2, batch vs auxiliary size, and captions.
>
> **Response:**
>
> - We rewrite the Table 2 caption as: “Top: near-OOD (CIFAR-100, TinyImageNet). Bottom: far-OOD (MNIST, SVHN, Textures, Places365). Entries are averages over near-OOD and far-OOD groups.”
> - Sec. 3.2 now states explicitly that the auxiliary set size is equal to the batch size $B$.
> - The minor caption and reference issues you noted have been corrected.

---

### Official Review · Reviewer_pypR · 2025-11-04

**Soundness:** 3
**Presentation:** 3
**Contribution:** 2
**Rating:** 4
**Confidence:** 4

**Summary:**

This paper addresses robustness issues in Evidential Deep Learning (EDL) for uncertainty quantification by investigating the impact of data augmentation strategies. The authors systematically study how pixel-space augmentations affect EDL's uncertainty estimates and propose LatentMix, a novel feature-space augmentation that interpolates latent representations across multiple network depths with layer-specific mixing severities. These severities directly regulate the EDL loss to shape aleatoric and epistemic uncertainty. Experiments show LatentMix improves out-of-distribution detection, better separates uncertainty types, and enhances calibration compared to pixel-space and single-layer mixing approaches.

**Strengths:**

- Overall, the paper is well written and easy to follow.
- The paper tackles an important problem.
- The proposed method is technically sound.
- The proposed method outperforms pixel-space and single-layer alternatives

**Weaknesses:**

- The method involves multiple components according to section 3.2 and 3.3. How sensitive is the proposed method with respect to these hyper-parameters?
- In general, the proposed method seems quite ad-hoc, most of which lacking theoretical justifications. It is not clear how well it would work outside the scenarios tested. Does this work across different architectures and domains?
- The authors of the paper use the OpenOOD benchmark. How does the method compare with other methods that use the OpenOOD benchmark? A thorough comparison against other methods in addition to EDL and mixup variants might be needed to demonstrated the effectiveness of the proposed method.
- In light of the fact that doing mixup in the latent space is studied extensively previously, it is not immediately clear to me if the contribution is sufficient from algorithmic perspective.

**Questions:**

- How does LatentMix compare to existing feature-space augmentations (MixFeat, ManifoldMixup)[1, 2]?
- How are mixing severities sampled? What's the distribution and hyperparameter sensitivity?

[1] "MixFeat: Mix Feature in Latent Space Learns Discriminative Space"
[2] "Manifold Mixup: Better Representations by Interpolating Hidden States"

---

> ### Author Response · Authors · 2025-11-19
> **Response on Robustness, Generality, and Design Choices**
>
> Thanks for the detailed and constructive feedback on robustness, generality, and positioning of our method.
>
> ### R1.1 Sensitivity to hyperparameters (Secs. 3.2-3.3)
>
> **Concern:** EviMix has multiple components and hyperparameters, robustness is unclear.
>
> **Response:**
>
> - All hyperparameters are listed in App. B.7. We add a sensitivity study on CIFAR-10 for the main quantities: severity cap $\kappa$, $\lambda_a$, $\lambda_e$, and the auxiliary/cross-class schedules (App. C).
> - Varying $\lambda_a,\lambda_e \in \{0.5,1.0,2.0\}$ and $\kappa \in \{0.5,0.75,1.0\}$ changes CIFAR-10 OOD AUROC by $\pm[0.3-2.8]\%$ and CIFAR-10-C ECE by $\pm[0.02-0.05]$, suggesting a broad robust plateau rather than a fragile configuration.
>
> **Clarification:** Sec. 3.3 states that our default values are chosen from this empirically stable region.
>
> ---
>
> ### R1.2 “Ad-hoc” design and generalization across architectures
>
> **Concern:** The design seems heuristic, and generality beyond tested setups is unclear.
>
> **Response:**
>
> - EviMix only assumes access to intermediate features and is therefore architecture-agnostic. To illustrate this beyond ResNet/VGG, we additionally report results (Sec. 4.2, Tab. 5) on:
>   - WideResNet-28-10 [1] on CIFAR-10
>   - ViT [2] backbone on ImageNet-200
>
> | Backbone             | Effect of EviMix vs. EDL+PixMix           |
> |----------------------|-------------------------------------------|
> | WideResNet-28-10 [1] | AUROC $+5.01\%$, ECE $-0.017$             |
> | ViT [2]              | AUROC $+4.9\%$, ECE $-0.012$              |
>
> These results follow the trends observed for our main backbones (Sec. 4.2), indicating that the evidence-mixing scheme transfers across architectures.
>
> **Clarification:** Sec. 4.2 includes a subsection “Generalization across architectures” summarizing Tab. 5 and noting that the same mixing mechanism is compatible with convolutional and transformer backbones.
>
> [1] “Wide Residual Networks”
> [2] “An Image is Worth 16x16 Words: Transformers for Image Recognition at Scale”
>
> ---
>
> ### R1.3 Comparison within OpenOOD / additional baselines
>
> **Concern:** Need more thorough comparison with methods using the OpenOOD benchmark.
>
> **Response:**
>
> - We already compare to several OpenOOD-style UQ methods (MC Dropout, KL-PN, RKL-PN, PostNet, I-EDL, DAEDL) in Table 3.
> - We clarify in Sec. 4.1 that our focus is on uncertainty-aware models and that we adopt the OpenOOD CIFAR-10 and ImageNet-200 protocols as standardized evaluation settings within which we study evidential uncertainty (aleatoric/epistemic).
>
> **Clarification:** Sec. 4.1 documents the OpenOOD protocols we use and explains that our primary goal is to understand how EviMix shapes evidential uncertainty rather than optimize leaderboard scores.
>
> ---
>
> ### R1.4 Contribution vs. existing feature-space mixup (ManifoldMixup)
>
> **Concern:** Latent mixing is well-studied, algorithmic novelty is unclear.
>
> **Response:**
>
> - Conceptually, EviMix aligns latent mixing with evidential modeling: the objective is to encourage a separation between aleatoric and epistemic evidence, rather than serve as a generic regularizer as in prior feature-space mixup methods.
> - Concretely, EviMix extends prior work along three axes (Sec. 3.2):
>   1. Depth-aware dual mixing (external early, cross-class late) instead of single-layer or uniform mixing.
>   2. Severity-aware evidential loss reweighting that explicitly couples augmentation strength to aleatoric vs. epistemic terms.
>   3. An EDL-specific curriculum (mixing cap schedule) designed to improve uncertainty quality (calibration and OOD) rather than accuracy alone.
> - We compare against ManifoldMixup in the same EDL setup (Sec. 4.2, Tab. 4). EviMix improves CIFAR-10 OOD AUROC by $5.3$\% and reduces CIFAR-10-C ECE by $0.021$ under matched backbones and training budgets.
> - Regarding MixFeat, we mention in Sec. 2 that it introduces additional architectural components and supervision not directly compatible with our evidential head without substantial re-implementation, so we treat it as a complementary feature-space method rather than a core baseline.
>
> **Clarification:** Secs. 2 and 3.2 now more clearly position EviMix as an uncertainty-aware evidence-mixing mechanism that builds on, but is conceptually distinct from, generic feature-space mixup.
>
> ---
>
> ### R1.5 Severity sampling and sensitivity
>
> **Concern:** Clarify sampling distributions and their effect.
>
> **Response:**
>
> - Sec. 3.2 spells out the severity sampling used in EviMix (auxiliary vs. cross-class, caps, schedules).
> - The effect of the 0.5 factor and the schedules is analyzed in a unified ablation study on CIFAR-10/ResNet-18 (Sec. 4.3, Tab. 6). Removing the 0.5 factor slightly worsens calibration and slows convergence.
>
> **Clarification:** The “Severity sampling” paragraph in Sec. 3.2 points directly to Tab. 6 for empirical evidence that our sampling choices act as practical stabilizers.

---

### Author Response · Authors · 2025-11-28
**Review-Focused Revision Summary**

- We thank all reviewers for their constructive feedback. Their comments helped us clarify the main goal of this work: EviMix is not intended as yet another generic mixup-style data augmentation, but as an uncertainty-aware evidence-mixing scheme. By combining evidential deep learning with carefully designed feature-space mixing, EviMix aims to encourage a meaningful separation between aleatoric and epistemic components, rather than only improving accuracy or raw OOD scores.

- The core methodology, model, and main conclusions remain unchanged. We reorganize the paper around:
  - methodological clarity
  - robustness and sensitivity
  - fairness with stronger baselines
  - clearer empirical validation

- Concretely, in response to reviewers’ comments, we:
  - (i) add hyperparameter sensitivity for main EviMix knobs (κ schedule, (λ_a, λ_e), factor 0.5). We clarify severity sampling (Uniform/Bernoulli with caps). This addresses R1.1, R1.5, R3.4, R4.2.
  - (ii) extend architecture coverage (WideResNet-28-10, ViT-Small/16). This addresses R1.2, R2.2, R2.3.
  - (iii) expand baselines with (CE, ensemble n=5, MSP scoring). We include recent mixing methods (LayerMix) in additiion to feature-mixing method (ManifoldMixup). We also update related work via recent mixup surveys. We also indicate why a method like MixFeat could not be established as baseline because of architectural change overhead. This addresses R1.3, R1.4, R3.4, R4.1.
  - (iv) enforce a matched 150-epoch training budget for all methods. We add training curves and 75 vs. 150 comparisons to show baseline saturation while EviMix continues improving. This addresses R3.1, R4.3.
  - (v) clarify external data exposure: PixMix and EviMix use the same public mixing set. We include an ID-only auxiliary severity variant to show gains are not from privileged data. This addresses R3.2, R4.3.
  - (vi) refine disentanglement wording to avoid claiming perfect UQ separation. We include a cross-task swap test to evaluate functional specialization between aleatoric (corruption) and epistemic (OOD) signals and link it to severity-loss coupling ablation. This addresses R3.3, R4.2.
  - (vii) shorten and improve structure, captions, naming, and remove repeated EDL definitions. This addresses R2.1, R2.4, R3.6.
  - (viii) report training-time overhead and peak GPU memory increase, while emphasizing unchanged inference-time cost. This addresses R3.5, R4.3.

---

### Meta-Review · Area_Chair_SjSW · 2025-12-20

**Summary:**

The reviews for this paper are mixed. This paper was reviewed by four experts in the field and received 1 Reject (2), and 4 Marginal Reject (4).

This article focuses on robustness and uncertainty estimation in deep learning based on evidential learning (EDL) techniques and examines how data augmentation strategies influence uncertainty quality. The authors introduce LatentMix, a feature space augmentation framework that interpolates latent representations across multiple network layers, extending ideas from pixel space methods such as MixUp, PixMix, and AugMix. LatentMix uses both interclass mixing within a patch and mixing with an external image set, with layer-specific mixing levels that explicitly regulate the EDL loss to shape random and epistemic uncertainty. Evaluated on image classification benchmarks, including CIFAR-10 and ImageNet variants using ResNet architectures, the proposed method consistently improves uncertainty calibration, out-of-distribution detection, and robustness to corruptions compared to standard EDL benchmarks and pixel space augmentation approaches.

Yet, based on the reviews, I side with the reviewers recommending rejection. The authors are encouraged to carefully consider the reviewers’ comments, to improve the paper for submission elsewhere.

**Reviewer Concerns:**

The reviewers agree that the article addresses an important issue and that the proposed LatentMix method is technically sound and empirically promising in the context of evaluation. However, the reviewers raised a number of significant concerns:
1. Limited novelty and positioning: While LatentMix extends mixup-style augmentation to feature space and across multiple layers, reviewers noted that latent/feature-space mixing has been extensively studied in prior work (e.g., Manifold Mixup, MixFeat, LayerMix).
2. Heuristic design and lack of theoretical grounding: Many fundamental assertions, such as the relationship between mixing depth and the distinction between aleatoric uncertainty and epistemic uncertainty, are largely heuristic and have not been directly validated. Several reviewers pointed out that these assertions are repeated several times, but have not been rigorously tested by targeted experiments or diagnostics.
3. Potentially unfair experimental comparisons: Multiple reviewers raised concerns about fairness in the evaluation. In particular, LatentMix is trained for substantially more epochs than baselines, appears to use corrupted or auxiliary data not available to competing methods, and employs dynamic training schedules that may not be applied consistently across baselines.
4. Incomplete evaluation and limited scope: The experiments are restricted to image classification with relatively standard CNN architectures, with no exploration of other modalities, tasks (e.g., regression), or modern architectures such as ViTs.
5. Presentation and clarity issues: While some reviewers found the article readable, others noted significant problems with its organization, presentation of results, missing references to tables/figures, and ambiguity in the details of the experimental protocol, all of which undermine its reproducibility and clarity.

**Reviewer Scores:**

I think that after the rebuttal, their score would not have increased, especially if they had participated in the discussion. Personally, I find the way the experiments are presented a little strange. The authors carefully chose which baseline to put in which table. Tables 1, 2, 3, 4, 5, 9, 10, and 12 concern a small CNN architecture, and the baselines are only mixed strategies. Table 6 reports the delta of the WideResnet, ViT-Small, and Resnet 50 results. Tables 7 and 8 are ablations, but the architecture they refer to is unclear. Table 11 shows sensitivity on Resnet18 and Cifar10. Finally, Table 14 shows different uncertainty techniques. We have at least CE and deep ensembles. Then when the reviewers asked for more architecture or baselines the authors claimed during the rebuttal that the demand is already satisfied, but no. Because the other architecture or baseline required were assessed in a particular setting not with all the other tables. First of all, I think it might be interesting to compare many more techniques, especially ensemble techniques, Batch Ensembles, MIMO, Mask Ensemble, Packed Ensembles, BNN, McDroupout, and also have a clear presentation. There are other strategies that mix strategies. In addition, Table 14 shows a really strange accuracy on Cifar 10 Resnet 18. Usually, the number of epochs should be around 74 and the accuracy around 94 [1], not 83 as stated in the article. I also share the reviewer's concerns regarding the experimental sections, and the author's response has not alleviated them. I spent some time reading the article, and according to the authors, they have corrected it. I find the corrections to be far from sufficient. I recommend spending more time addressing the reviewer's comments. The experimental section should also be given special attention in the new version of the article. 99% of the tables deal only with evidential or mixed strategies. If the author does not wish to compare to other uncertainty quantification strategies, this should be explained. I recommend that the authors take the time to redo all the experiments and rewrite the experimental section. Although the idea is interesting, there is still a lot of work to be done before the article can be accepted.


[1]  Laurent, O., et al. "PACKED-ENSEMBLES FOR EFFICIENT UNCERTAINTY ESTIMATION." 11th International Conference on Learning Representations, ICLR 2023. International Conference on Learning Representations, ICLR, 2023.

---

### Decision · Program_Chairs · 2026-01-26

Reject